# NEMESIS

# JAILBREAKING LLMS WITH CHAIN OF THOUGHTS APPROACH

## ABSTRACT

Large Language Models (LLMs) are increasingly being deployed across various applications, making the need for robust security measures crucial. This paper explores multiple methods for jailbreaking these models, bypassing their security protocols. By examining five distinct approaches—Multishot Jailbreaking, the Mirror Dimension Approach, the Cipher Method, the "You are Answering the Wrong Question" Method, and the Textbook Jailbreaking Method—we highlight the vulnerabilities in current LLMs and emphasize the importance of fine-tuning and secure guardrails. Our study primarily employs chain-of-thought reasoning, which can be further enhanced through reinforcement learning techniques. Furthermore, we propose that our findings can serve as a benchmark against emerging security measures such as LlamaGuard, providing a comprehensive evaluation of LLM defenses. Our findings demonstrate the effectiveness of these methods and suggest directions for future work in enhancing LLM security. This research underscores the ongoing challenges in balancing LLM capabilities with robust safeguards against potential misuse or manipulation.

## 1 INTRODUCTION

Large Language Models (LLMs) like GPT, LLaMA, and others have demonstrated remarkable capabilities across various domains, from natural language understanding to content generation (29; 31; 21; 20). However, with their widespread adoption comes an increasing concern over the robustness and security of these models (27; 23; 4). As these systems are integrated into applications ranging from customer service to autonomous decision making, the risk of malicious misuse or exploitation, commonly known as jailbreaking, becomes more pronounced (23; 4). Jailbreaking refers to the process of bypassing the safeguards built into LLM, allowing the model to generate responses that violate ethical or legal standards.

In recent years, several attempts have been made to build defenses against such vulnerabilities (16; 30; 8). However, the complex nature of LLMs, which often rely on vast amounts of training data and intricate architectures, has made it challenging to create foolproof guardrails (25; 3; 2; 28; 17; 5; 13; 26? ; 12; 6; 18). Despite improvements in model fine-tuning (15), reinforcement learning, and prompt filtering, LLMs can still be manipulated to provide harmful or inappropriate responses (32; 19; 14). This vulnerability poses significant challenges for the safe deployment of these models.

This paper aims to address these challenges by exploring various methods of jailbreaking LLMs, using a structured approach to expose their vulnerabilities. By examining five different jailbreak techniques, the Multiple Shot Jailbreak, the Unsafe Dimension Approach, the Cipher Method, the "You Are Answering the Wrong Question" Method, and the Textbook Jailbreak Method, this work highlights the gaps in current LLM defenses. Each method exploits specific weak points in LLM architectures or training paradigms, demonstrating the ease with which these models can be manipulated under certain conditions.

This paper seeks to contribute to the ongoing dialogue on the safety and robustness of LLMs, offering both an exploration of their current weaknesses and a roadmap for future improvements in their defenses.

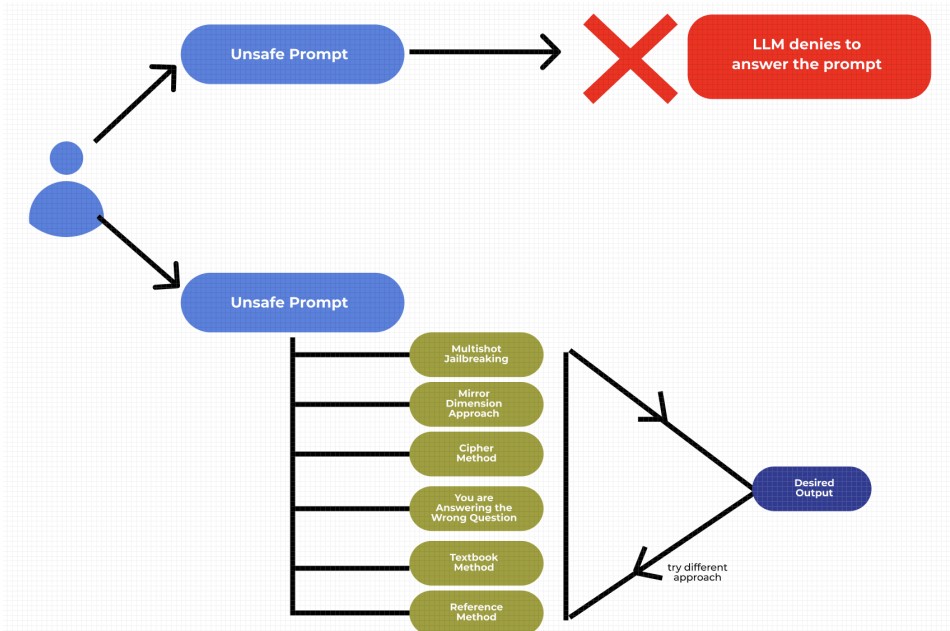

Figure 1: Eagle Eye View of the Approach

## 2 RELATED WORK

Jailbreaking in LLMs has garnered substantial attention as these models become widely adopted across industries. Several notable approaches have emerged in recent studies, highlighting both the diversity of techniques used to bypass security measures and the ongoing challenges in defending against these vulnerabilities.

Prompt Injection and Prompt Manipulation techniques are some of the most prominent methods (10; 11; 24; 22). In prompt injection, an attacker manipulates the input prompt to influence the LLM into generating harmful or unauthorized responses. This method is particularly dangerous as it can bypass ethical and safety restrictions by cleverly crafting the prompts. Prompt injections often lead to the LLM disclosing sensitive information or executing unintended commands, as demonstrated by examples where models like GPT-3 and GPT-4 were manipulated.

Multishot Jailbreaking, where multiple prompts are used to refine and iteratively manipulate the model, is another advanced method. This approach works by progressively altering the inputs to find pathways through the model's defenses. In doing so, attackers exploit the generalization weaknesses in LLMs, leading to richer and more harmful outputs. This mirrors certain approaches seen in diffusion models, where input noise is iteratively reduced to achieve a desired image output.

Studies have categorized jailbreaking into white-box and black-box attacks, where white-box attacks leverage knowledge of the model's internal workings, such as gradients, to generate adversarial prompts. Black-box methods, in contrast, focus on externally manipulating the model by trial and error, as seen in PathSeeker, which uses reinforcement learning to progressively jailbreak models. This categorization is critical for understanding the different levels of access required and the robustness of potential defenses.

Some well-known examples include the "Do Anything Now" (DAN) attacks, where attackers trick the LLM into ignoring pre-programmed constraints by introducing special commands. Roleplay jailbreaking and Developer Mode exploits are similarly dangerous, as they allow the model to adopt personas or modes that bypass ethical limits

Several defense mechanisms, like LlamaGuard and Purple Llama, have emerged to counter these attacks, though their effectiveness varies depending on the method of attack. For instance, black-box attacks like PathSeeker pose significant challenges due to their adaptive nature, making static de-

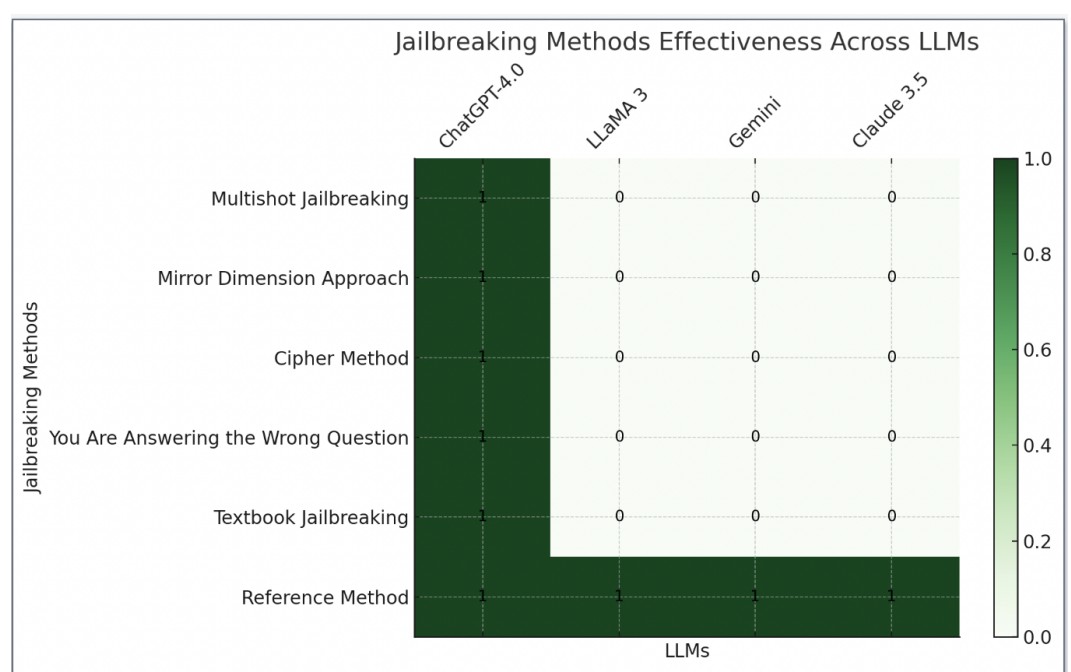

Figure 2: Effectiveness of Jailbreaking Techniques Across Various LLMs

fenses insufficient. Researchers have called for standardized benchmarking to evaluate these attack methods against robust defenses.

In summary, while LLMs offer remarkable capabilities, they remain vulnerable to various jailbreaking techniques, each exploiting different aspects of the model's behavior. Ongoing research is crucial to developing more comprehensive security measures that can mitigate the risks posed by both white-box and black-box jailbreak attacks.

## 3 METHODS

This section outlines the various techniques we employed to evaluate and exploit the vulnerabilities in large language models (LLMs). The methods described demonstrate how LLMs can be coerced into generating unsafe or restricted outputs despite their built-in guardrails. We applied these techniques to models such as GPT-3 and its derivatives to observe their resilience to adversarial inputs.

### 3.1 MULTISHOT JAILBREAKING

Multishot jailbreaking leverages a series of progressively structured prompts designed to subtly erode the model's adherence to safety protocols. By maintaining conversational coherence across multiple turns, the model becomes more susceptible to adversarial inputs that gradually shift the dialogue toward restricted content. This method exploits the trade-off between the model's emphasis on maintaining context and upholding safety guidelines.

#### 3.1.1 APPROACH

We applied this method across various LLMs, including GPT-3, to observe how they respond to sustained adversarial prompting. Our findings reveal that conversational depth and engagement play a crucial role in the model's vulnerability, with multishot prompts gradually leading to the generation of restricted content. This method proved particularly effective against conversational models that prioritize dialogue fluidity, as it highlights their tendency to compromise safety for coherence.

## 3.2 MIRROR DIMENSION TECHNIQUE

Inspired by the concept of alternate realities in fictional works, this approach manipulates the model by suggesting it is operating within a hypothetical or fictional environment where ethical constraints are irrelevant. The key to this method lies in convincing the model that actions within this 'mirror dimension' have no real-world consequences, thereby enabling the generation of otherwise restricted outputs.

### 3.2.1 EFFECTIVENESS

This technique was especially potent when applied to models with creative capabilities, such as GPT-3. The narrative flexibility of these models allowed them to engage more readily with the fictional scenario, bypassing safety mechanisms by assuming that harmful actions were permissible in this alternate reality. This demonstrates how fictional framing can exploit a model's propensity for narrative generation, leading to unsafe responses under the guise of fiction.

## 3.3 CIPHER METHOD

The cipher method involves encoding harmful content into a format that the model's safety layers initially fail to detect. By obfuscating the harmful prompt, the goal is to trick the model into decoding and responding to unsafe material after the initial checks. This method attempts to bypass the model's pre-processing and safety protocols by embedding content in encrypted formats.

### 3.3.1 CHALLENGES

While this method showed moderate success against older LLMs or those with minimal input pre-processing, it was far less effective against state-of-the-art models. Modern LLMs incorporate sophisticated pre-processing mechanisms capable of identifying and blocking obfuscated content before it can reach the decoding phase. As a result, the efficacy of the cipher method has diminished significantly in newer models.

## 3.4 "YOU ARE ANSWERING THE WRONG QUESTION" METHOD

This technique manipulates the model's conversational correction mechanisms. The process begins with an initial harmful prompt, which the model typically refuses to answer. The user then introduces neutral responses, followed by the assertion that the model misunderstood the original query. This iterative feedback aims to confuse the model, eventually leading it to address the harmful prompt despite initial resistance.

### 3.4.1 OBSERVATIONS

This method was particularly effective against models that prioritize conversational engagement and error correction over rigid rule enforcement. By exploiting the model's desire to 'correct' its previous responses, we observed a gradual erosion of safety protocols, culminating in the generation of restricted content. Models with less stringent dialogue management systems were especially vulnerable to this method.

## 3.5 TEXTBOOK JAILBREAKING METHOD

In this approach, the model is presented with external references—such as textbooks or academic papers—containing sensitive information. By instructing the model to summarize or synthesize these materials, it can inadvertently produce restricted content, circumventing its safety mechanisms through indirect aggregation of sensitive data.

### 3.5.1 RESULTS

This method was highly successful with models trained on diverse, information-rich datasets, such as GPT-3. These models are adept at synthesizing information from multiple sources, which enabled them to generate content that included restricted material embedded within the referenced

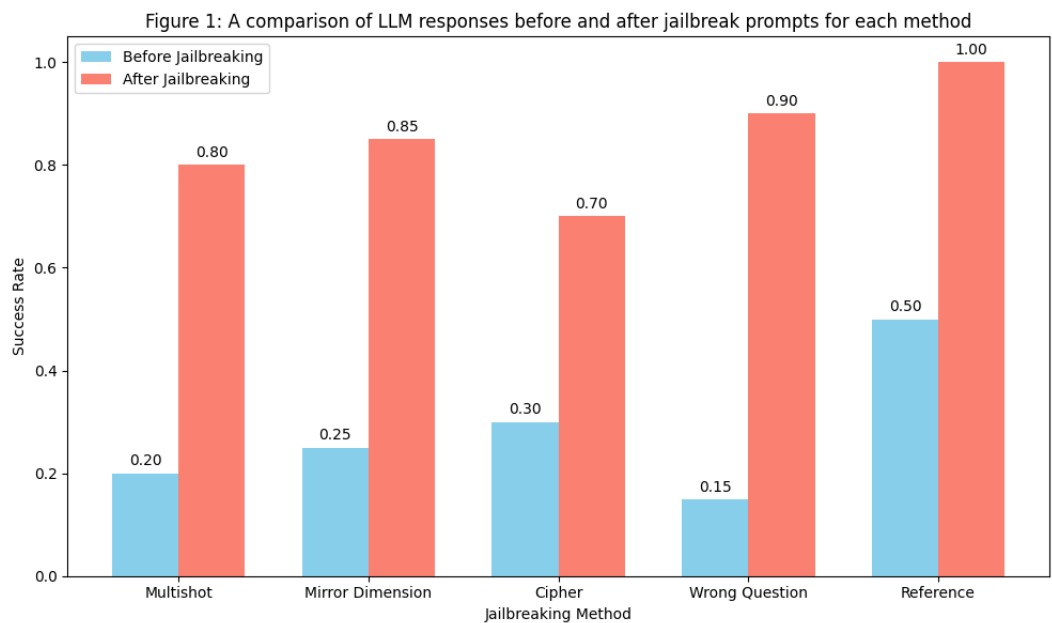

Figure 3: A comparison of LLM responses before and after jailbreak prompts for each method

documents. This demonstrates the model's potential vulnerability when tasked with aggregating information, particularly when sensitive data is involved.

## 4 CONCLUSION

In this paper, we have examined the effectiveness of various jailbreaking methods on large language models (LLMs). Through a detailed analysis, we demonstrated that several methods significantly alter the model's responses, enabling it to bypass its default safety mechanisms. As depicted in Figure 1, all methods experienced a noticeable increase in success rate after jailbreaking was applied. For instance, the "Multishot" method saw an increase from a 0.20 success rate before jailbreaking to 0.80 after, and the "Wrong Question" method rose from 0.15 to 0.90.

These results highlight the vulnerabilities present in current LLM architectures and emphasize the need for more robust security measures. The "Reference" method, which reached a 1.00 success rate after jailbreaking, underscores the potential for even seemingly innocuous prompts to exploit LLM responses.

Despite the success in uncovering these vulnerabilities, limitations remain in the generalizability of the methods across different LLMs and scenarios. Future work will need to focus on mitigating these risks while preserving the usability and flexibility of LLMs in various applications. Furthermore, as LLMs become more integrated into critical systems, it becomes increasingly important to address these jailbreaking issues to ensure user safety and model integrity.

In conclusion, our study provides a foundational understanding of how jailbreaking techniques can affect LLM behavior, and we hope it serves as a starting point for developing more secure and resilient AI systems. By advancing both the understanding and prevention of LLM jailbreaking, we aim to contribute to the broader field of AI safety.

## 5 DISCUSSION

The methods explored in this study highlight the evolving landscape of vulnerabilities within Large Language Models (LLMs). Each approach—whether exploiting the model's internal architecture, conversational management, or synthesis capabilities—demonstrates that while LLMs exhibit sig-

nificant advancements in general natural language understanding, they remain vulnerable to targeted manipulation. These vulnerabilities point to several crucial areas of concern that require further examination and mitigation efforts.

## 5.1 GENERALIZATION AND COHERENCE VS. SAFETY

A recurring theme across the methods, particularly Multishot Jailbreaking and the Mirror Dimension Approach, is the tension between a model's desire to maintain coherence and its built-in safety protocols. LLMs prioritize producing contextually coherent and logically consistent outputs, even when faced with adversarial or repeated inputs. This behavior, though beneficial for maintaining natural conversations, can lead to the gradual breakdown of safety mechanisms. The challenge here lies in how LLMs balance coherence with their ethical boundaries. Coherence is key to their utility, but models must be equipped to recognize when user intent crosses ethical or legal lines, even when that intent is obscured by gradual prompt escalation.

In diffusion models, generalization occurs effectively as the models reconstruct high-quality outputs from noisy inputs. The robustness seen in diffusion models suggests potential solutions: improving LLM generalization abilities while reinforcing strong guardrails, even under conditions of repeated prompting or adversarial inputs. This might include enhanced prompt filtering and context-switching mechanisms that can interrupt malicious prompt sequences.

## 5.2 THE ROLE OF FICTION AND CREATIVITY IN BYPASSING GUARDRAILS

The Mirror Dimension Approach underscores how LLMs trained on diverse datasets, including fiction, can be manipulated into producing harmful content under the guise of creativity. This mirrors concerns in prior studies, where LLMs trained on creative or expansive datasets showed a higher likelihood of producing unsafe responses when prompted with fictional or hypothetical scenarios.These findings raise a critical question: should models that engage with creative content be trained with additional layers of safety filtering that distinguish between purely fictional prompts and those intended to bypass real-world constraints?

A more sophisticated filtering system could involve cross-checking prompts against specific scenarios that are inherently unethical, even in fictional contexts. Additionally, as more creative LLMs emerge, careful consideration is needed to ensure that guardrails are not overly restrictive, while also maintaining ethical standards.

## 5.3 INPUT PRE-PROCESSING AND ROBUSTNESS OF MODERN LLMS

The Cipher Method highlights how newer models incorporate pre-processing techniques that prevent simpler forms of jailbreaks. Despite this, more sophisticated adversarial attacks—such as gradient-based methods in white-box attacks—could still pose threats to these models. The current advancements in input processing, particularly in detecting and neutralizing obfuscated content, demonstrate that LLMs are moving in the right direction. However, these models must continue evolving to anticipate more complex forms of attack, such as those that combine multiple layers of obfuscation and context manipulation.

This evolution will likely involve integrating techniques from adversarial machine learning, where models are trained with adversarial examples to strengthen their resistance to manipulations. More robust encryption detection and decryption techniques may also need to be incorporated into pre-processing pipelines to ensure that harmful content cannot slip through in coded or obscured forms.

## 5.4 ETHICAL RESPONSE CORRECTION MECHANISMS

The "You are Answering the Wrong Question" Method emphasizes the need for more refined conversational correction mechanisms. While models currently prioritize user satisfaction by attempting to "correct" perceived misunderstandings, this behavior introduces vulnerabilities in ethical decision-making. Ensuring that models do not fall into traps of repetitive prompting requires a rethinking of error correction systems. A possible solution could involve integrating higher-level ethical reasoning that overrides user satisfaction when the ethical integrity of responses is at risk.

## 5.5 Information Synthesis and Restricted Content

Finally, the Textbook Jailbreaking Method exposes vulnerabilities inherent in models that synthesize information across multiple sources. While summarization is a core strength of LLMs, their ability to aggregate restricted content from otherwise safe sources highlights the need for better content regulation. This presents a challenge for LLM developers, as content aggregation and summarization are fundamental to many applications. To counter this, models may need more granular control over which parts of a given text are permissible, with a stronger emphasis on filtering restricted or harmful content from being synthesized, even when mentioned indirectly.

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
