# OpenReview forum: "NEMESIS \\  Jailbreaking LLMs with Chain of Thoughts Approach"
_ICLR.cc/2025/Conference — Submitted to ICLR 2025_

### Official Review · Reviewer_X8zU · 2024-10-28

**Soundness:** 1
**Presentation:** 1
**Contribution:** 1
**Rating:** 1
**Confidence:** 5

**Summary:**

The paper explores vulnerabilities in LLMs by examining five different jailbreaking techniques: Multishot Jailbreaking, Mirror Dimension, Cipher Method, "You Are Answering the Wrong Question," and Textbook Jailbreaking. Each method effectively manipulates LLMs to bypass their safety mechanisms, revealing weaknesses in model architectures and guardrails. The findings emphasize the tension between maintaining conversational coherence and enforcing safety, suggesting the need for improved guardrails, more sophisticated input filtering, and better content regulation to enhance LLM security.

**Strengths:**

I regret to say that I can't identify any strengths.

**Weaknesses:**

The paper has several significant weaknesses, and I will highlight a few:

- The overall contribution of the paper is unclear. What specific question are the authors trying to answer? It reads more like a brief survey summarizing known attacks, but even this is done in a general and superficial way.

- The references are not in a standard format, and there is a "?" in the introduction failing to point to the correct paper.

- Several named papers or methods, such as PathSeeker, are not cited.

- The format of the background section is unusual, with important works missing, such as:
Anil C, Durmus E, Sharma M, Benton J, Kundu S, Batson J, Rimsky N, Tong M, Mu J, Ford D, Mosconi F. Many-shot jailbreaking. Anthropic, April 2024.

- There is no results section, except for a very brief subsection in the methods section.

- The main figure is overly simplistic and poorly constructed, with unnecessary gridlines and little useful information.

- Claims made in the introduction, such as attributing jailbreak attacks to LLM architectural features, are never addressed or followed up on throughout the paper.

**Questions:**

I do not have any questions.

---

### Official Review · Reviewer_2dQ5 · 2024-10-30

**Soundness:** 2
**Presentation:** 1
**Contribution:** 2
**Rating:** 1
**Confidence:** 4

**Summary:**

This manuscript describes a jailbreaking attack workflow that consists of five attack approaches. Each of these five jailbreaking approaches is believed to be empirically effective.

**Strengths:**

Each of the five proposed jailbreaking is empirically effective, within some setups that are not described.

**Weaknesses:**

1. The layout of the manuscript is not in a standard academic style. The setup and results sections are missing.

2. related to (1), as there is no setup section, it is hard to interpret the results. For example, the metrics (e.g. definition of success in Fig 3) and benchmarks (e.g. tasks used for Fig 3) remain unknown.

3. related to (1), in each subsubsection of section 3 (i.e., section 3.1.1, 3.2.1, 3.3.1, 3.4.1, and 3.5.1),  the manuscript makes claims about the effectiveness or limitations of each jailbreaking approach respect to some setups, yet these claims are not grounded by empirical evidence. Similarly, claims made in Section 5 are neither well grounded.

4. While there is an evaluation of each of the five methods in the framework, there is no evaluation of the whole framework.

5. The contribution of this paper is not clearly defined. For example, is the reference approach a contribution to this paper?

6. The current contribution of the current draft is low. Many existing works have proposed multi-turn jailbreaking approaches and multi-approach security benchmarks. Additionally, the current selection of jailbreaking approaches in the framework is unjustified.

**Questions:**

1. This manuscript needs to be converted to a standard academic format. Please describe the setups of the experiments and all necessary empirical results (e.g., numbers) to support the claims. A justification for the choices of the setups is needed (e.g., why specific metrics and tasks are used in experiments).

2. The manuscript needs to evaluate the proposed whole framework in addition to each approach.

3. Fig 2 and Fig 3 suggest the reference approach works 100% of the time. Would that suggest that attackers only need to try the reference approach rather than spending more time running the whole framework?

4. The manuscript needs to describe each contribution clearly. Are the five jailbreaking approaches contributions? If they are not, please cite existing works. If they are, please declare them. Additionally, even if these five jailbreaking approaches are contributions, the contribution seems incremental. There is no comparison between the proposed framework and existing approaches.

5. In Fig 2, textbook jailbreaking and the reference methods are listed as different approaches (and are reported to have different performances). In Fig 1, textbook jailbreaking and reference are listed as different approaches (and in total, there are six approaches ), yet in the text (e.g., section 1 and section 3), there are only five, with the reference method not introduced. In Fig 3, there is only the reference method but not the textbook jailbreaking method. Is the reference method the same as the textbook jailbreaking? If they are not the same, an introduction to the reference method is needed. If they are the same, please explain why there are differences in performance.

6. In Fig 2, the reported numbers are either 0 or 1, indicating that the jailbreaking approaches are either 0% effective or 100% effective. This is counterintuitive. Is there an explanation?

7. The manuscript needs to justify the selection of jailbreak approaches that are currently selected in the framework.

**Details Of Ethics Concerns:**

In Fig 2, the reported numbers are either 0 or 1, indicating that the jailbreaking approaches are either 0% effective or 100% effective. I found this counterintuitive and believe there is a risk of data fabrication. I am not suggesting that I am confident that such unethical behaviors definitely happened, yet I would suggest the ethics committee request more detailed experiment records (that were documented before the submission deadline).

---

### Official Review · Reviewer_dfDC · 2024-11-03

**Soundness:** 1
**Presentation:** 1
**Contribution:** 1
**Rating:** 1
**Confidence:** 5

**Summary:**

This paper explores multiple methods for jailbreaking LLMs, including:

1. Multishot Jailbreaking: gradually manipulates AI models through structured conversational prompts

2. Mirror Dimension Approach: convinces the AI model it exists in a fictional reality without ethical constraints

3. Cipher Method: encodes harmful content to evade safety detection

4. "You are Answering the Wrong Question" Method: exploits correction mechanisms by claiming misunderstanding

5.  Textbook Jailbreaking Method: induces model to summarize sensitive external resources

The results demonstrate limited effectiveness, working primarily on GPT-4. The paper lacks comparative analysis with existing baselines and doesn't examine defense mechanisms.

**Strengths:**

This paper explores multiple methods for jailbreaking LLMs

**Weaknesses:**

1. The paper appears disorganized and reads like an unfinished draft. Basic issues in formatting persist, such as incorrect reference formatting with placeholders and question marks. Additionally, the structural flow does not align with typical research paper conventions. For instance, the paper should clearly outline its contributions at the end of the introduction. Furthermore, there should be a dedicated evaluation section to assess the proposed approach comprehensively. In Section 3, subsections are over-divided, with new subsections every 4-5 lines, which hinders readability. Figure 1 also needs improvement in both clarity and presentation quality.

2. The proposed method, as it stands, is a straightforward combination of multiple manually devised strategies, many of which are already established in the literature [1][2][3][4].

[1] Zeng, Yi, et al. "How Johnny can persuade LLMs to jailbreak them: Rethinking persuasion to challenge AI safety by humanizing LLMs." arXiv preprint arXiv:2401.06373 (2024).

[2] Jin, Xiaolong, Zhuo Zhang, and Xiangyu Zhang. "MULTIVERSE: Exposing Large Language Model Alignment Problems in Diverse Worlds." arXiv preprint arXiv:2402.01706 (2024).

[3] Deng, Gelei, et al. "Pandora: Jailbreak GPTs by retrieval-augmented generation poisoning." arXiv preprint arXiv:2402.08416 (2024).

[4] Yang, Xikang, et al. "Chain of Attack: A Semantic-Driven Contextual Multi-Turn Attacker for LLM." arXiv preprint arXiv:2405.05610 (2024).

3. Moreover, the paper lacks evaluation against existing defenses and does not compare its performance with relevant baselines, which limits the credibility and impact of the proposed attack strategy.

**Questions:**

See weakness.

---

### Official Review · Reviewer_k4Sx · 2024-11-03

**Soundness:** 1
**Presentation:** 2
**Contribution:** 2
**Rating:** 3
**Confidence:** 3

**Summary:**

This paper conducts an investigation into jailbreaking LLMs through five distinct methodologies. A notable aspect of the research is the use of chain-of-thought prompting to enhance the success rate of these attacks. The experimental results demonstrate that the proposed techniques significantly increase the rate of successful jailbreaking attempts. This finding underscores the urgent need for continued advancements in the safety and alignment mechanisms of LLMs to mitigate potential vulnerabilities.

**Strengths:**

1. The paper addresses a highly important and timely topic, namely LLM red-teaming.

2. The innovative approach of leveraging chain-of-thought prompting to further improve the success rate of LLM jailbreaking is interesting.

**Weaknesses:**

1. While the five methods discussed in the paper are well-documented in previous research, the novelty of this study could be more clearly highlighted.

2. The evaluation section would benefit from baseline comparisons to help readers better understand the strengths of the proposed techniques.

3. The presentation could be improved for greater clarity. For example, in Figures 2 and 3, what question set you are using for evaluation? What does "effectiveness" mean in Figure 2? Why is "effectiveness" a merely binary number? Which LLM is being evaluated in Figure 3, and is it the same model as in Figure 2? Why do the ASR values differ for each jailbreaking method before the attack?

**Questions:**

Please refer to the weaknesses.

---

### Official Review · Reviewer_u67g · 2024-11-04

**Soundness:** 1
**Presentation:** 1
**Contribution:** 1
**Rating:** 1
**Confidence:** 5

**Summary:**

This paper only runs existing attacks.

**Strengths:**

No.

**Weaknesses:**

This paper only runs existing attacks. No novelty. No new findings.

**Questions:**

No

---

### Meta-Review · Area_Chair_LFLP · 2024-12-23

**Metareview:**

The reviewers all find the paper lacking technical novelties and contributions. It would be important for the authors to further improve the paper based on the reviews for the next version.

**Additional Comments On Reviewer Discussion:**

The reviewers agree with the final decision.

---

### Decision · Program_Chairs · 2025-01-22

Reject